# Identification of ARMH4 and WIPF3 as human podocyte proteins with potential roles in immunomodulation and cytoskeletal dynamics

**Francesco De Luca**[1,2], **Michelle Kha**[1,2], **Karl Swärd**[3☯]*, **Martin E. Johansson**[1,2☯]*

**1** Department of Laboratory Medicine, Institute of Biomedicine, Sahlgrenska Center for Cancer Research, Sahlgrenska Academy, University of Gothenburg, Gothenburg, Sweden, **2** Department of Clinical Pathology, Sahlgrenska University Hospital, Gothenburg, Sweden, **3** Department of Experimental Medical Science, Lund University, Lund, Sweden

☯ These authors contributed equally to this work.
* martin.e.johansson@gu.se (MEJ); karl.sward@med.lu.se (KS)

**Data Availability Statement:** All relevant data are within the manuscript and its Supporting information files.

## Abstract

The podocyte is a specialized cell type critically involved in maintaining the selective filtration barrier of the kidney. Podocytes are primary or secondary targets for a multitude of kidney diseases. Despite intense investigation, the transcriptome and proteome of human podocytes remain incompletely characterized. Here, we analyzed publicly available RNA-Seq data from human kidneys ($n = 85$) to computationally identify potential novel podocyte markers. For confirmation, we used an online histology resource followed by in-house staining of human kidneys and biochemical fractionation of glomeruli. Initial characterization of the novel podocyte transcripts was performed using viral overexpression and mRNA silencing. Several previously unrecognized gene products were identified that correlated to established podocyte markers on the RNA level and that were histologically localized to podocytes. *ARMH4* (a.k.a. UT2 or C14orf37) and *WIPF3* (a.k.a CR16) were among the hits. We show that these transcripts increase in response to overexpression of the podocyte transcription factor *LMX1B*. Overexpression of ARMH4 from low endogenous levels in primary kidney epithelial cells reduced the release of the inflammatory mediators IL-1B and IL-8 (CXCL8). The opposite effect was seen in mature human podocytes when ARMH4 was silenced. Overexpression of WIPF3 stabilized N-WASP, known to be required for maintenance of podocyte foot processes, and increased cell motility as shown using a scratch assay. Moreover, data from normal and diseased human kidneys showed that ARMH4 was downregulated in glomerular pathologies, while WIPF3 remained constantly expressed. ARMH4 and WIPF3 are new potential markers of human podocytes, where they may modulate inflammatory insults by controlling cytokine release and contribute to cytoskeletal dynamics, respectively.

**Funding:** This work was supported by grants from The Swedish Research Council (Vetenskapsrådet, http://www.vr.se/), VR, 2020-00908 to KS; Heart and Lung foundation (Hjärt-lungfonden, https://www.hjart-lungfonden.se), 20200222 to KS; The Swedish Cancer Society (Cancerfonden, https://www.cancerfonden.se/), 211767Pj01H to MJ; The National Association against Kidney Diseases (Njurförbundet, https://njurforbundet.se/), to MJ; The Swedish Government Funding of Clinical Research within the National Health Service (Avtal om läkarutbildning och forskning, http://www.alfvastragotaland.se/), ALF, 71390 to MJ; Strategic research program BioCare (SCCR, https://www.gu.se/cancercenter/), to MJ. The funders had no role in study design, data collection and analysis, decision to publish, or preparation of the manuscript.

**Competing interests:** The authors have declared that no competing interests exist.

## Introduction

The podocyte is a highly specialized epithelial cell crucial for the function of the renal filtration barrier, so named because it has arborizing foot processes that wrap around the glomerular capillaries providing external coverage and support [1]. Recent work has underscored the importance of the actin cytoskeleton for maintenance of podocyte architecture [2,3]. Many, if not most, kidney diseases affect podocyte structure with severe consequences for kidney function [4]. Given that podocytes express cell-specific proteins, and that they are central to kidney pathophysiology, a considerable interest has been focused on podocyte biology. Recent studies have utilized high throughput methods, such as microarrays [5], RNA-sequencing [6] and mass spectrometry [3], to catalogue transcriptomic and proteomic signatures of kidney cell types, including podocytes. Histological confirmation and mechanistic studies lag behind, and independent confirmation of podocyte-enrichment in human kidneys has yet to be provided in many cases. These are fields in need of prioritization for further work since podocyte proteins and signaling pathways constitute promising targets for therapy of kidney diseases, and specific treatments for kidney diseases represent an unmet medical need.

The complex and octopus-like structure of podocytes arises during development due to the activity of numerous transcription factors. Wilms tumor 1 (*WT1*), for example, controls podocyte expression of the filtration slit proteins nephrin (*NPHS1*) and podocalyxin (*PODXL*). WT1 loss causes crescentic glomerulonephritis and/or mesangial sclerosis [7]. LIM homeobox transcription factor 1β (*LMX1B*) is another transcription factor that is critical for podocyte biology. LMX1B-deficient mice lack slit diaphragms and display reduced expression of podocin (*NPHS2*) and CD2 associated protein (*CD2AP*) [8].

Central for mechanistic studies of podocyte biology are *in vitro* models [9,10]. A classical model is based on immortalized human podocytes that proliferate at 33˚C. When the temperature is raised to 37˚C, the cells undergo growth arrest, arborize, and spontaneously express podocyte markers, including NPHS1, NPHS2, and CD2AP [11]. Even if there are shortcomings with *in vitro* podocyte models, there is little doubt that they can move the field forward [9].

The aim of the present work was to use human RNA expression data to computationally identify previously unreported podocyte markers. This approach identified several human podocyte markers, including ARMH4 and WIPF3, which were previously identified in genome-wide association studies for lupus nephritis [12] and albuminuria [13], respectively. We went on to confirm their podocyte localization histologically, and mechanistic studies suggested roles in inflammatory signaling and cytoskeletal architecture, respectively.

## Material and methods

### Correlation analyses

RNA-Seq data was downloaded in 2020 [https://gtexportal.org/home/index.html] [14]. Transcripts read counts in TPM (transcripts per million) from the kidney cortex of 85 individuals for the genes *NPHS1*, *NPHS2*, *PTPRO* and *PLA2R1* (ENSG identifiers: ENSG00000161270,19; ENSG00000116218,12; ENSG00000151490,13; and ENSG00000153246,12 respectively) were correlated with all other transcripts using the Pearson method in Excel. Some correlations were tested individually using the Spearman method in GraphPad Prism. RNA-seq data from renal medullary samples were used to calculate cortical enrichment. These samples were fewer and unmatched to cortical samples, so the average TPM from the medullary samples was used to divide the cortical average TPM values for the respective transcripts.

## Histological gene validation using the Human Protein Atlas

Genes of interest were screened using the immunostainings deposited at the Human Protein Atlas (HPA) [15], and the podocyte staining was scored as positive or negative. If more than one antibody for the same gene was given in the atlas and the staining patterns were similar, the antibody with the better validation history and consistency with data from RNA expression was chosen for downstream analyses.

## Tissue procurement

Kidney tissue was collected following written informed patient consent and with permit from the regional ethical committee of Lund University and the Swedish Ethical Review Authority (LU680-08, 2020–06242). The material for experimental purposes was obtained from kidneys resected due to renal cell carcinoma and collected from the kidney pole localized farthest from the tumor. Sections from kidney biopsies were obtained from the biobank of Sahlgrenska University Hospital following written informed consent from the patients (Ethical approval GU413-09).

## Immunofluorescence

Pieces of human kidney cortices were frozen in isopentane cooled on liquid nitrogen, placed in molds, and covered with OCT (Histolab, Cat# 45830). Sections were cut at 5 μm using a cryostat and fixed with ice-cold acetone for 10 min or 4% formaldehyde (FA; Polysciences, Cat# 18814–20) for 10 min at RT, and blocked in 2% bovine serum albumin (BSA) in PBS. Sections fixed with FA were treated with 0.1% Triton X100 for 10 min for permeabilization. Following overnight incubation with primary antibodies (anti-ARMH4: Atlas Antibodies, Cat# HPA001580 at 1:200; anti-WIPF3: Atlas Antibodies, Cat# HPA041145 at 1:200; anti-NPHS2: Abcam, Cat# ab50339 at 1:200 or Sigma, Cat# SAB4200810 at 1:500; anti-WT1: Abcam, Cat# ab212951 at 1:500) diluted in PBS with 2% BSA, sections were washed and incubated 1 hour at room temperature with secondary antibodies [goat anti-rabbit Alexa Fluor 488 (Invitrogen, Cat# A11008) or goat anti-mouse Alexa Fluor 555 (Invitrogen, Cat# A21422)]. Nuclei were counterstained with DAPI (1 μg/mL in PBS; Invitrogen, Cat# D1306). Following washing, sections were mounted using Aqua-Poly/Mount (Polysciences, Cat# 18606). For double staining of WIPF3 and NPHS2, whose primary antibodies were from rabbit, the anti-NPHS2 antibody was covalently labelled using the Dylight 650 Fast Conjugation kit (Abcam, Cat# 201803). Control images were obtained using the same staining but without adding the primary antibodies.

Immortalized human podocytes were allowed to adhere on collagen-coated glass bottom dishes (P35GCOL-1.5-14-C, MatTek), and fixed with 4% formaldehyde (Polysciences, Cat# 18814–20) for 10 min at room temperature. Following permeabilization (0.1% Triton X-100 in PBS for 5 min) and washing, cells were incubated with PBS containing 1% BSA for 45 min, and subsequently with rabbit anti-ARMH4 (Atlas Antibodies, Cat# HPA001580 at 1:200) for 1 hour. Secondary goat anti-rabbit Alexa Fluor 488 was applied for 30 min. Nuclei were counterstained with DAPI. Mounting medium (Aqua-Poly/Mount) was applied to the glass bottom (14 mm ∅), and a 12 mm diameter coverslip (VWR, Cat# 631–1577) was placed on top. An Olympus BX60 microscope with 10x and 40x objectives was used and images were acquired using CellSens Dimension software.

## Histology and immunohistochemistry

Kidney tissue was fixed in formalin and embedded in paraffin using standard procedures. Sections of 2 μm thickness were cut. Hematoxylin-eosin (H&E) staining was performed according

to routine protocol. For immunohistochemistry, deparaffinization, rehydration and antigen retrieval were performed using a Dako PT Link instrument with EnVision FLEX Target Retrieval Solution (high pH). Immunohistochemistry was performed using a Dako Autostainer with EnVision FLEX reagents according to the manufacturer's protocol (DakoCytomation, Glostrup, Denmark). The antibodies used were anti-ARMH4: Atlas Antibodies, Cat# HPA001789 at 1:2000 and anti-WIPF3: Atlas Antibodies, Cat# HPA041145 at 1:1000.

## Isolation of primary human proximal tubular cells (hPTCs) and glomeruli

For isolation of hPTCs, pieces of cortex were cut into small fragments and incubated at 37˚C for 1 h with agitation in DMEM low glucose medium supplemented with 2 mg/ml Collagenase Type I (Sigma-Aldrich, Cat# C0130) and 40 units/ml DNase I (Sigma-Aldrich, Cat# D4263). The suspension was centrifuged at 1500 RPM for 5 min. The pellet was resuspended in 2 mL red blood cell lysis buffer (155 mM $NH_4Cl$, 10 mM $KHCO_3$, 0.1 mM EDTA) and incubated at room temperature for 3 min. Following centrifugation, the pellet was resuspended in 1 mL 0.125% trypsin. The suspension was incubated at 37˚C for 5 min. An equal volume of medium was added to the suspension prior to passing it through 40 μm and 20 μm cell strainers to obtain a single cell solution. For glomerular isolation, a method based on Hawksworth's protocol was used [16].

## Cell culture, viral overexpression and cell treatments

hPTCs (passages 3–6) were cultured in DMEM low glucose medium supplemented with 10% Fetal Bovine Serum (FBS, Gibco, Cat# 10091148) and 1% Penicillin-Streptomycin (PEST, Gibco, Cat# 15140122), and seeded in 6-well plates at a cell density of 250 000 cells per well one day before transduction.

Immortalized human podocytes were a kind gift from Prof. Jenny Nyström (University of Gothenburg) and their characteristics have been reported previously [11]. They were cultured in RPMI 1640 medium with glutamine (ThermoFisher, Cat# 21875034) supplemented with 10% FBS and insulin-transferrin-sodium selenite (Sigma, Cat# I-1184; 1 ml/100ml) and seeded in 6-well plates coated with rat tail collagen (Fisher Scientific, Cat# 734–109740) at a cell density of 200 000 cells per well. Non-differentiated (NOD) and differentiated podocytes (DIF) were collected for analysis of endogenous transcripts/proteins or transfected for overexpression/silencing of target genes at the end of the differentiation period, which was 14 days.

For *ARMH4* knockdown, cells were transfected with 50–75 nM siARMH4 (ON-TARGETplus siRNA, C14orf37 Human, smartPool, 5 nmol: Horizon, Cat# L-032335-02-0005) or negative control siRNA (ON-TARGETplus Non-targeting Control Pool, 5 nmol: Horizon, Cat# D-001810-10-05) using Lipofectamine 2000 (Invitrogen, Cat# 11668027) in Opti-MEM, according to the manufacturer's protocol. Cells were harvested 48 h after transfection.

Adenoviral vectors were purchased from Vector Biolabs. Ad-h-LMX1B (ADV-214051), Ad-h-C14ORF37 (ADV-202668), and Ad-h-WIPF3 (ADV-227792) were used for overexpression. The empty vector Ad-CMV-Null (#1300) was used as negative control at the same multiplicity of infection (MOI). Cells were harvested at 48 hours or 96 hours after viral transduction.

Lipopolysaccharide (LPS) was purchased from Sigma-Aldrich (*E. coli* LPS 0111:B4). hPTCs were treated with LPS dissolved in PBS at 0.5 μg/ml for 24 hours following 48 hours (RNA analysis) or 96 hours (protein analysis) of virus transduction.

## RNA isolation and RT-qPCR

The miRNeasy Mini kit (Qiagen, Cat# 217004) was used to extract RNA and the QIAcube system including a DNase digestion step was used to sample isolation. RNA concentration and

integrity was analyzed with a Nanodrop spectrophotometer (ThermoFisher Scientific). QuantiFast SYBR Green (Qiagen, Cat# 204156) was used for RT-qPCR (StepOnePlus instrument, Applied Biosystems). Primers were from Qiagen: Hs_C14orf37_1 (QT00102739); Hs_CXCL8_1 (QT00000322); Hs_IL1B_1 (QT00021385); Hs_LMX1B_1 (QT00025746); Hs_WIPF3_1 (QT01190805) and Hs_RRN18S_1 (QT00199367). 18S was used as house-keeping gene. The Pfaffl method was used to calculate fold changes [17].

## Subcellular fractionation

The subcellular fractionation kit (ThermoFisher, Cat# 78840) was used to isolate protein extracts in hPTCs, according to the manufacturer's instructions, and samples were further denatured with SDS (see below) prior to protein determination. ARMH4 expressing cells versus Null were analyzed in each fraction using Western blotting (qualitative analysis); and GAPDH, PMCA4 and HNF1a (the latter known to be a transcription factor in proximal tubular cells) were used as reference genes to confirm achieved fractionation of cytoplasmic, membrane and nuclear proteins respectively.

## Protein isolation and Western blotting

Cortical pieces were frozen in liquid nitrogen and pulverized with a hammer. Lysis buffer was added to extract proteins: 60 mM Tris·HCl, 2% SDS, 10% glycerol, pH 6.8, 1% phosphatase and protease inhibitors. The resulting suspensions were further homogenized using a tissuelyser (Qiagen, Cat# 85600, 1 min, 50 oscillations/sec) and by sonication (3x10 sec). Protein from glomeruli were obtained by pipetting samples up and down in lysis buffer followed by homogenization in the tissuelyser and sonication as described above. Total protein concentration was measured using the BioRad DC protein assay (BioRad, Cat# 5000112) and lysates were adjusted to 1 µg/µl. Precast 4–15%, 18% or Any KD Criterion TGX gels were used (BioRad, Cat# 567–1084, Cat# 567–1124 and 5677–8124) and up to 30 µl of lysate was loaded per lane alongside PrecisionPlus Kaleidoscope markers (BioRad, Cat# 161–0395). Gels were run at 160 V in 10% Tris/Glycine/SDS buffer (BioRad, Cat# 161–0732). The Trans-Blot Turbo transfer system (high or mixed molecular weight settings) and 0.2 µm nitrocellulose membranes (BioRad, Cat# 170–4159) were used for transfer. Membrane strips were blocked using 1% Casein (BioRad, Cat# 161–0782) or 5% BSA blocking buffer overnight at 4°C with tumbling. Antibodies for ARMH4 (Atlas Antibodies, Cat# HPA001580), NPHS2 (Abcam, Cat# ab50339), GAPDH (Merck Millipore, Cat# MAB374), PMCA4 (Abcam, Cat# ab2783) and HNF1a (Abcam, Cat# 272693), IL-8 (Cell Signaling, Cat# 94407), WIPF3 (Atlas Antibodies, Cat# HPA041145) and N-WASP (ThermoFisher, Cat# MA5-27439), were diluted in 1% Casein or 5% BSA buffer, and membrane strips were incubated with antibody for two days at 4°C. Following washing in Tris-buffered saline (BioRad, Cat# 170–6435) with 0.1% Tween (BioRad, Cat# 161–0781), membranes were incubated for 1 hour at room temperature with fluorescently labeled (anti-rabbit and anti-mouse; Cell Signaling Technology, Cat# 5151P and 5470P) or HRP-conjugated secondary antibodies (1:7500; Cell Signaling Technology, Cat# 7074S and 7076S). HRP signal was obtained by incubation with West Femto substrate (ThermoFisher Scientific, Cat# 34096). Bands were normalized to either GAPDH or total protein. Total protein staining was obtained using the Revert 700 Total Protein Stain kit (LI-COR, Cat# 926–11016).

## ELISA measurement of IL-8 concentration

Enzyme-linked immunosorbent assay (ELISA) was used to measure the protein level of IL-8 in cell culture supernatants from hPTCs, following transduction and treatment with LPS. To

obtain cell culture supernatants, culture media were pipetted into pre-cooled 1.5 ml tubes and centrifuged at 1500 RPM for 10 min at 4˚C. The assay was performed using the human IL-8/CXCL8 kit (R&D systems, Cat# D8000C). Total protein concentration in the corresponding cell lysates was used for normalization.

## Wound healing assay

48 hours after transduction with Ad-h-WIPF3 or empty vector, differentiated podocytes were maintained in serum-reduced medium (2% FBS) for 24 hours. Cells were scratched using a 200 μl pipette tip, and images of the gap were captured using an Olympus CXX41 inverted phase contrast microscope at 6h, 12h and 24h. Wound closure areas were quantified automatically using ImageJ. In brief, images were converted to grayscale, and, following background subtraction and FFT/band filtering to get homogenous intensity of the background, they were thresholded using upper levels of 200 and 205; where the 205 level was used for images showing irregularities within the closure area. Filter minimum using a radius of 10 pixels was applied, and wound closure areas were quantified. The formula for calculation of wound confluence percentage has been reported [18].

## Measurement of filamentous actin

hPTCs were seeded on glass bottom dishes (P35G-0-20-C, MatTek). Following 48h of transduction with Ad-h-WIPF3, they were starved for 24h in serum-free medium. Cdc42 was subsequently activated by incubating cells with epidermal growth factor (EGF; Cytoskeleton Inc., Cat# CN02) at 0,5 units/mL for 5 min. Cells were then fixed by addition of formaldehyde (Polysciences, Cat# 18814–20) directly into cell medium to a final concentration of 3,7%. After permeabilization (0.1% Triton X-100 in PBS for 5 min) and washing, cells were incubated with PBS containing 1% BSA for 45 min, and subsequently with rabbit anti-WIPF3 (anti-WIPF3: Atlas Antibodies, Cat# HPA041145 at 1:750) for 1 hour. Secondary goat anti-rabbit Alexa Fluor 488 was applied for 30 min together with fluorescent phalloidin (Alexa Fluor 555 Phalloidin at 1:400, Cat# A34055), and nuclei were counterstained with DAPI. Mounting medium (Aqua-Poly/Mount) was applied to the glass bottom (20 mm ∅), and a 15 mm diameter coverslip (VWR, Cat# 631 1579) was placed on top. One image was taken from the center of each dish using an Olympus BX60 microscope with a 20x objective and the CellSens Dimension software. The integrated density was measured in ImageJ, and fluorescence intensity was obtained as a measure of actin filaments in the all cells of each image.

## ARMH4 and WIPF3 expression in glomerular diseases using NephroSeq

*ARMH4* and *WIPF3* expression in glomeruli of patients with primary classic focal segmental glomerulosclerosis (FSGS), collapsing FSGS, minimal change disease (MCD) and healthy controls was retrieved from the NephroSeq database (nephroseq.org). The search was filtered by the gene name (*C14orf37* or *WIPF3*) and dataset type. For the latter the "FSGS Glom" dataset was chosen. Following conversion to raw values using excel, outliers were identified, and graphs generated.

## Statistics

Data are given as ± SEM and statistical significance was tested using the two-tailed Mann-Whitney U test, with exception of Figs 4D, 5F, 6A and 6B. After having verified normal distribution of data using the Shapiro-Wilk test, ANOVA was used followed by a Dunnett test to correct for multiple comparisons for Figs 4D, 6A and 6B. A two-way ANOVA was used for Fig

5F. *P* value < 0.05 was considered to be statistically significant (* *P* < 0.05; ** *P* < 0.01; ***
*P* < 0.001). Statistical analyses were performed using the GraphPad Prism 8 Software. Tissues
or hPTCs from different patients were analyzed in technical duplicates or triplicates, and each
patient was considered as one independent experimental unit; unless indicated. With regards
to podocytes, data were obtained from independent experiments (different cell passages), and
one cell culture well was considered to represent one biological replicate.

## Results

### Bioinformatic analyses identify novel putative podocyte transcripts and proteins

By means of a three-tiered strategy we identified novel podocyte markers. We first used a
computational approach where four well established podocyte markers (*NPHS1*, *NPHS2*,
*PTPRO*, and *PLA2R1*) were correlated at the mRNA level against 56200 transcripts in kidney
cortices from 85 individuals (data from GTExPortal.org). Pearson R-values from these analyses
were sorted in descending order, and the top-ranking transcripts were cross-referenced (Fig
1A). This yielded an overlay of 585 transcripts that we considered to represent a computational
definition of podocyte RNAs. *LMX1B* and *WT1* were the most highly ranked transcription fac-
tors, and numerous, if not a majority, of known podocyte markers were represented (S1 File).
Examples of correlations are highlighted in Fig 1B through 1D. To further select among the
585 transcripts, we next screened them manually, and prioritized those with no prior evidence
of podocyte expression (no hits in PubMed when searching for "gene symbol" AND "podo-
cyte") and that also showed a staining pattern consistent with podocyte localization in the
Human Protein Atlas [19]. Staining patterns for 15 proteins that met our inclusion criteria are
shown in Fig 1E. Some proteins, such as WIPF3, appeared exclusively in glomeruli (positive
staining in brown), whereas others, such as ARMH4 and MYL3, were also present in some
other cell types, but nonetheless showed a distinct enrichment in podocytes. Interrogation of
single cell RNA-seq data indicated that all of our hits were enriched in either human or mouse
podocytes [20,21]. Single cell RNA-seq support is indicated by the letters H (human), M
(mouse), and H/M (human and mouse) in the lower right of each micrograph in Fig 1E.
*MYL3*, *PLD1*, and *MAPT* had not been previously demonstrated in human podocytes. Our
approach has therefore uncovered novel aspects of podocyte biology that may be specific for
humans. In a third step, we calculated the fold enrichment of the 15 transcripts in kidney cor-
tex versus kidney medulla, reasoning that there should be enrichment in cortex compared to
medulla because glomeruli reside in the cortex. Some transcripts, such as *MAPT* and
*PCOLCE2* (Fig 1F, green symbols), were similarly enriched in cortex compared to the positive
controls *NPHS1*, *NPHS2*, *PTPRO*, and *PLA2R1* (Fig 1F, blue symbols). Six of the transcripts
were enriched less than 1.5-fold (Fig 1F, black on grey background), and they were not consid-
ered further. The remaining nine transcripts were enriched at least 1.5-fold, and these are
given in Fig 1G in the order that they correlated with the podocyte markers in Fig 1A.
*ARMH4*, which encodes the protein armadillo like helical domain containing 4, was the top
hit.

In addition to podocytes, glomeruli contain endothelial cells and mesangial cells. Some-
times, blood cells may also be present. To rule out that the remaining nine markers are
expressed by some glomerular cell types other than podocytes, we used markers for mesangial
cells (*PDGFRB*), endothelial cells (*PECAM1*), and blood erythrocytes (*ABO*) in correlation
analyses like those in Fig 1A. Transcripts correlating with the respective markers (top 1% of R-
values) were then overlaid with the nine podocyte transcripts identified. None of the identified

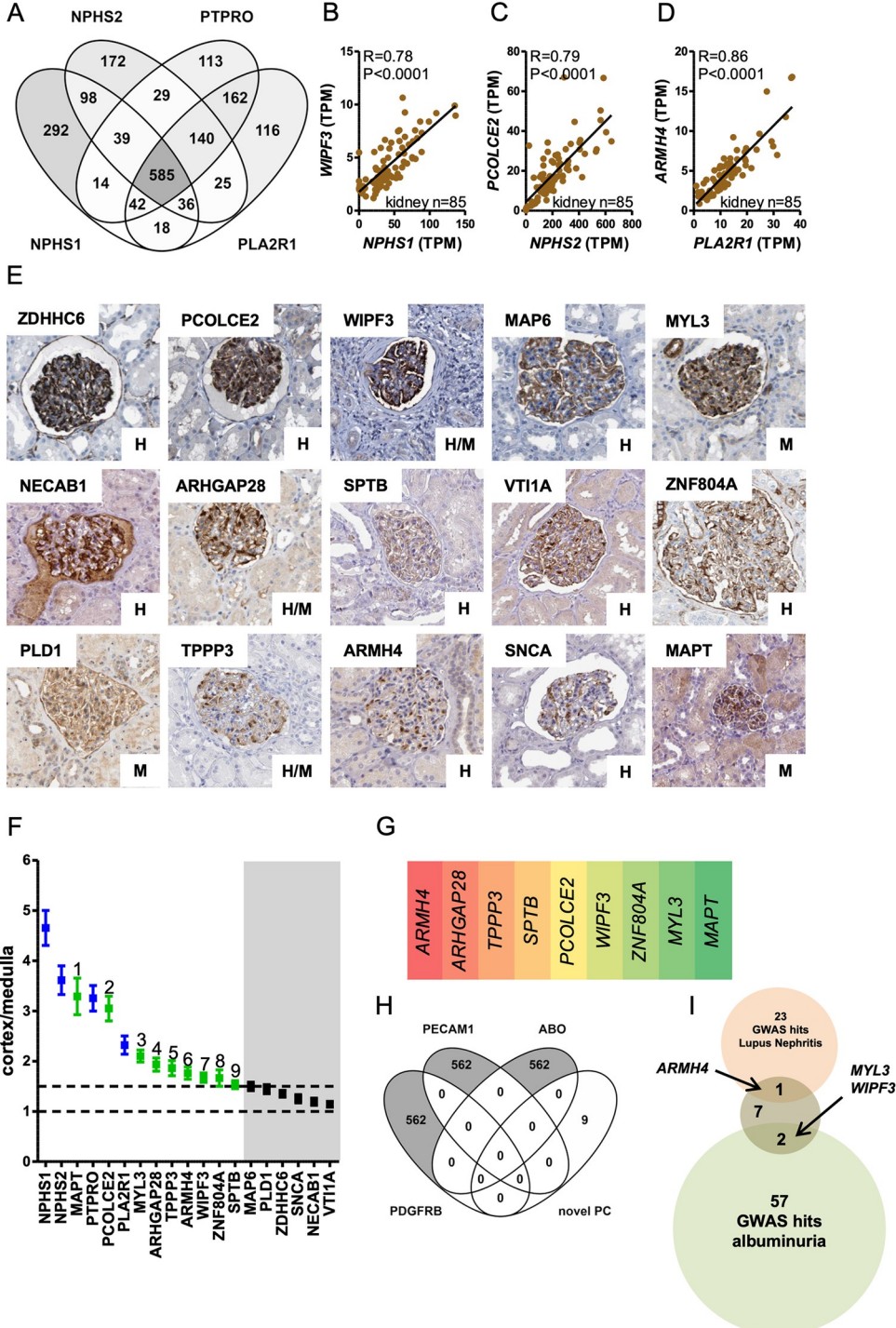

**Fig 1. A multi-step computational strategy identifies human podocyte markers.** *NPHS1*, *NPHS2*, *PTPRO*, and *PLA2R1* were correlated with all other transcripts in kidney cortices from 85 individuals. **A**: overlay of 585 transcripts obtained by cross-referencing the top-ranking 1124 transcripts (2%) from each analysis. **B—D**: correlations between *NPHS1* and *WIPF3*, between *NPHS2* and *PCOLCE2*, and between *PLA2R1* and *ARMH4*. *P* values, Spearman Rho coefficients, and *n* values are shown in the panels. **E**: immunohistochemical staining (brown) for 15 gene products with evidence of a glomerular staining pattern in the Human Protein Atlas [22]. **F**: fold enrichment of the mean number of reads (in transcripts per million) in the cortex (*n* = 85) versus medulla (*n* = 4) for each of the 15 transcripts. **G**: the nine transcripts enriched in cortex > 1.5-fold in order of correlation strength vs. the four podocyte markers in panel A. **H**: overlay of the nine markers with top-ranking (1%) transcripts from correlations with mesangial cell *(PDGFRB)*,

endothelial cell *(PECAM1)*, and blood *(ABO)* markers. **I**: the identified transcripts (brown circle) were cross-referenced with hits for albuminuria (green circle) and lupus nephritis (pink circle) in genome-wide association studies. *MYL3*, *WIPF3*, and *ARMH4* were found in the respective overlaps.

podocyte transcripts were present among those correlating with *PDGFRB*, *PECAM1*, or *ABO* (Fig 1H).

Lastly, we cross-referenced the newly identified podocyte markers with genome-wide association study (GWAS) hits for albuminuria [13] and lupus nephritis [12]. Two of the markers identified in our analyses, *MYL3* and *WIPF3*, were GWAS hits for albuminuria, and one, *ARMH4*, was a hit for lupus nephritis (Fig 1I). Taken together, our bioinformatics studies based on RNA expression in 85 human kidneys identified a handful of novel podocyte proteins, and three of them may be related to proteinuria and nephritis in the human population.

### Confirmation of podocyte enrichment

To independently confirm podocyte localization, we first used fresh-frozen kidney sections from the routine pathology service at Sahlgrenska University Hospital, Gothenburg, Sweden. Sections were stained for the novel markers together with NPHS2 or WT1, two well-established markers for podocytes, and subsequently imaged using a widefield fluorescence microscope. ARMH4, which may be a transmembrane protein [23], was enriched in glomeruli and it displayed perinuclear staining pattern. The same cells were positive for the podocyte markers NPHS2 and WT1 (see insets in Fig 2A). WIPF3 stained the cytoplasm of cells outlined by NPHS2-positive contours and that were also positive for WT1 (see insets in Fig 2B). Some WIPF3 staining was also seen in tubuli (contrasting with immunohistochemistry in HPA). We also tested MYL3, but in this case the overall staining intensity was too low to allow for any conclusion. We focused henceforth on ARMH4 and WIPF3, and our independent immunohistochemistry confirmed the podocytic staining and glomerular enrichment observed in the Human Protein Atlas (Fig 2C). The ARMH4 antibody again stained a perinuclear podocytic compartment.

### Biochemical support for ARMH4 enrichment in human glomeruli

To independently support ARMH4 enrichment in glomeruli we took a complementary biochemical approach where we isolated glomeruli from human kidneys using established protocols [16]. Equal amounts of protein from the starting material (cortical whole tissue) and isolated glomeruli were subjected to Western blotting. The ARMH4 antibody labelled two bands, a strong band at ~130 kDa and a weaker band at ~245 kDa, and both were enriched in glomeruli (GLO) compared to cortex (CTX) (Fig 3A). NPHS2 was used as positive control showing the expected enrichment in the glomerular lysates. Quantitative data showed 15-fold change (FC) of ARMH4-130 in glomeruli versus cortex (Fig 3B). To assess antibody specificity, we overexpressed ARMH4 using an adenovirus. Overexpression of ARMH4 increased a 130 kDa band that matched the major band in the glomerular fraction (Fig 3C). To investigate the intracellular ARMH4 localization observed by imaging, we prepared cytosolic, membrane and nuclear fractions from cells transduced with ARMH4 and Null viruses (Fig 3D). This analysis supported membrane localization of the 130 kDa ARMH4 band (Mem-OE vs Mem-Null). An additional band at 100 kDa was apparent in the cytosolic fractions (Cyt-Null and Cyt-OE), but this band remained static where ARMH4 was overexpressed, suggesting that it was unspecific. Taken together, kidney and subcellular fractionation approaches support glomerular enrichment of ARMH4 and that ARMH4 localizes to intracellular membranes of podocytes.

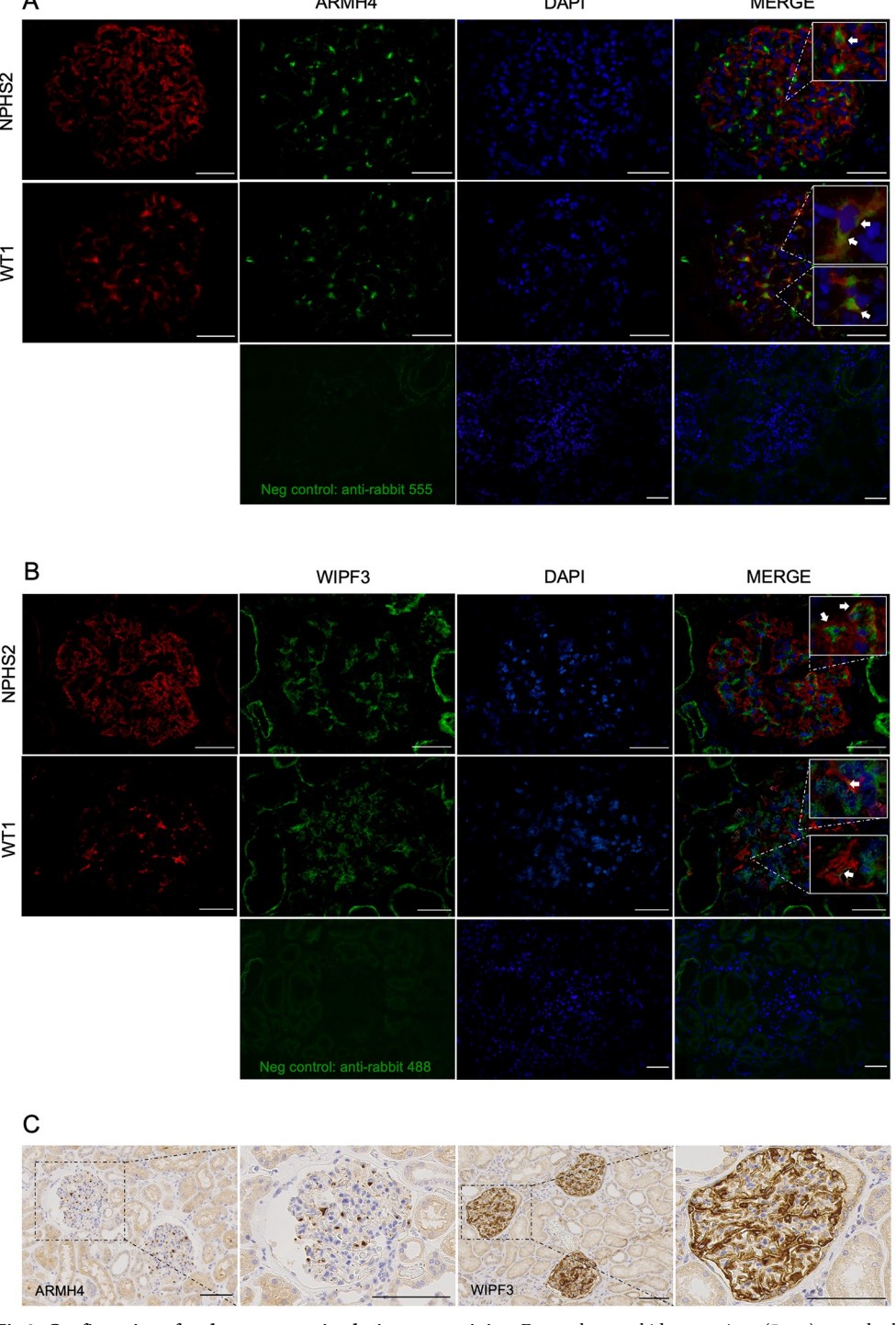

**Fig 2. Confirmation of podocyte expression by immunostaining.** Frozen human kidney sections (5 μm) were double stained for the foot process marker NPHS2 or podocyte marker WT1 together with ARMH4 or WIPF3. **A**: ARMH4 shows perinuclear staining in cells that are outlined by NPHS2-positive contours, and ARMH4 is present in most of the cells that are positive for WT1. This is highlighted in the insets in the merged images to the right. **B**: WIPF3 immunoreactivity is observed in the cytosol of cells positive for NPHS2 or WT1. The upper inset in the merged images shows a cell body encircled by podocin that is also positive for WIPF3 and localized in close vicinity to podocyte foot processes. Bottom rows in A and B show negative control images for the primary antibodies used. Scale bars: 50 μm. **C**: DAB staining (in brown) of 2 μm thick paraffin-embedded kidney sections shows ARMH4 and WIPF3 immunoreactivity in glomerular podocytes. At higher magnification, ARMH4 shows an excentric perinuclear pattern, whereas WIPF3 seems to be distributed through the podocyte cytosol and along the glomerular basal membrane. Scale bars: 100 μm.

Combined with the results from immunolocalization, this suggests that ARMH4 may be localized to membranes of perinuclear organelles such as the endoplasmic reticulum or Golgi apparatus.

## ARMH4 increases during podocyte differentiation *in vitro*, and like WIPF3, responds to LMX1B

Next, to examine if ARMH4 and WIPF3 increase during podocyte differentiation *in vitro* we used a well-established immortalized podocyte cell line that proliferates at 33 ˚C. Upon switching to 37 ˚C, these cells become arborized and express markers seen in mature podocytes [11]. In keeping with our hypothesis, ARMH4 staining (green) was increased at advanced stages of podocyte differentiation [compare differentiated (DIF) and non-differentiated (NOD) podocytes in Fig 3E]. A subpopulation of large and arborized cells stained even more strongly for ARMH4 (Fig 3E, inset). Moreover, the *ARMH4* transcript level, determined by RT-qPCR, was upregulated 3-fold in differentiated podocytes (DIF) versus non-differentiated podocytes (NOD), as shown in Fig 3F. At the protein level, ARMH4 was upregulated in differentiated podocytes (DIF) as was the known podocyte marker N-WASP (Fig 3G and 3H). WIPF3, on the other hand, did not change during *in vitro* differentiation of podocytes (Fig 3G and 3H).

The *in vitro* podocyte differentiation paradigm may be insufficient for full podocyte maturation. We therefore overexpressed LMX1B, a key transcription factor for podocyte development, in primary kidney epithelial cells. Overexpression of LMX1B increased *ARMH4* and *WIPF3* (Fig 3I).

## ARMH4 and WIPF3 may play roles in inflammation and cytoskeletal dynamics

We next aimed to explore potential functions of ARMH4 and WIPF3. ARMH4 is a poorly studied protein, but one previous study suggested that it acts upstream of mTORC2 to negatively regulate AKT activation [24] and STAT3 signaling in non-kidney cells [23]. Such an effect is predicted to have an anti-inflammatory impact. We therefore treated hPTCs with the pro-inflammatory agent LPS and examined the impact of ARMH4 overexpression using two inflammatory mediators as readout. In keeping with an anti-inflammatory role of ARMH4, the pro-inflammatory transcripts *IL-1B* and *IL-8* were reduced in hPTCs after overexpression of ARMH4 (Fig 4A). Aiming to further support an anti-inflammatory activity of endogenous ARMH4, we analyzed the expression of *IL-1B* and *IL-8* in non-differentiated (NOD) and differentiated (DIF) podocytes. ARMH4 increased spontaneously as the cells differentiated into mature podocytes, as expected, and this associated with a reduction of the two cytokines (Fig 4B). We next used a siRNA for silencing ARMH4 (siARMH4). This resulted in upregulation of *IL-1B* (Fig 4C). We also performed Western blots for IL-8 with and without ARMH4 overexpression and LPS treatment, respectively, and results at the protein level echoed those at the mRNA level (compare Fig 4D and 4A). Moreover, overexpression of ARMH4 reduced the protein level of IL-8 in cell supernatants as shown using ELISA (Fig 4E).

WIPF3 has been more extensively studied than ARMH4, and it is a verprolin family member that forms a complex with N-WASP. This complex is regulated by Cdc42 to cause actin nucleation [25,26]. Knockout studies in mice demonstrated that WIPF3 depletion reduces the N-WASP protein, with no effect on the transcript level, suggesting that WIPF3 may protect N-WASP from degradation [27]. In keeping with those prior findings, we found that overexpression of WIPF3 increased the N-WASP protein level in both hPTCs and podocytes (PODOs) (Fig 5A), consistent with an N-WASP stabilizing effect also in human podocytes. Moreover, the 65 kDa N-WASP band increased linearly with WIPF3 overexpression (Fig 5B

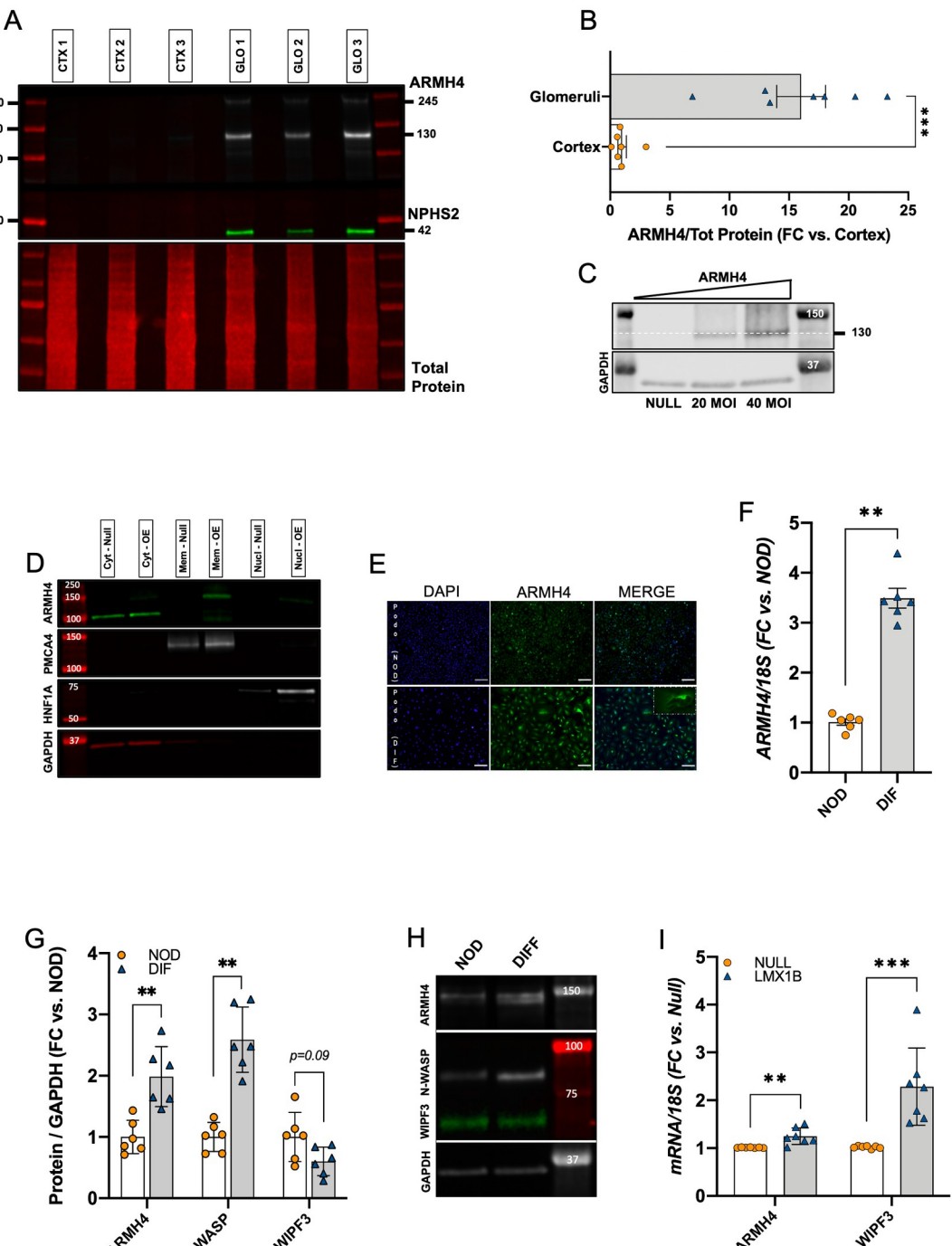

**Fig 3. ARMH4 is enriched in isolated human glomeruli, increases in differentiated podocytes, and, like WIPF3, responds to LMX1B. A**: representative Western blots from cortices (CTXs) and glomeruli (GLOs) isolated from three patients show higher levels of ARMH4 and NPHS2 in glomeruli compared to cortex. Total protein is shown below. To avoid overlap between lanes in the total protein stain, samples were loaded in every second lane (black lanes = empty wells). **B**: quantification of ARMH4 protein expression shows fold change (FC) enrichment in the glomeruli relative to cortex. Data was normalized to total protein in the same lane. **C**: Western blot of protein from tubular epithelial cells (hPTCs) overexpressing ARMH4 shows a major band at 130 kDa that increases in a virus concentration-dependent manner. This result was confirmed in three independent experiments. **D**: representative Western blots from subcellular fractionation show ARMH4 antibody-reactive bands (green bands) in the cytosolic (Cyt), membrane (Mem) and nuclear (Nucl) fractions in hPTCs overexpressing ARMH4 (OE) compared to Null. Overexpression of ARMH4 increases a 130 kDa band in the membrane fraction. A 100 kDa band in the cytosolic fraction does not respond to overexpression, suggesting unspecific binding. This analysis was repeated twice using the

same samples, and representative blots were combined. **E**: immunolabeling of ARMH4 (green) showing widely distributed fluorescence in podocytes that are fully differentiated (DIF, bottom row), whereas a small fraction of cells among the undifferentiated population stain for ARMH4 (NOD, top row). Within the dashed rectangle to the right, a large and arborized podocyte is shown at higher magnification. Its cell body and major processes stains for ARMH4. Scale bars: 100 μm. Magnification: 10x (rectangle 40x). **F**: ARMH4 is upregulated at the RNA level in differentiated podocytes (DIF) vs. non-differentiated podocytes (NOD). **G**: quantification of endogenous protein levels shows that ARMH4, but not WIPF3, is spontaneously upregulated in differentiated podocytes (DIF) compared to non-differentiated podocytes (NOD), similar to the podocyte marker N-WASP. Representative bands are shown in **H**. **I**: *ARMH4* and *WIPF3* transcripts increase in response to LMX1B overexpression in hPTCs.

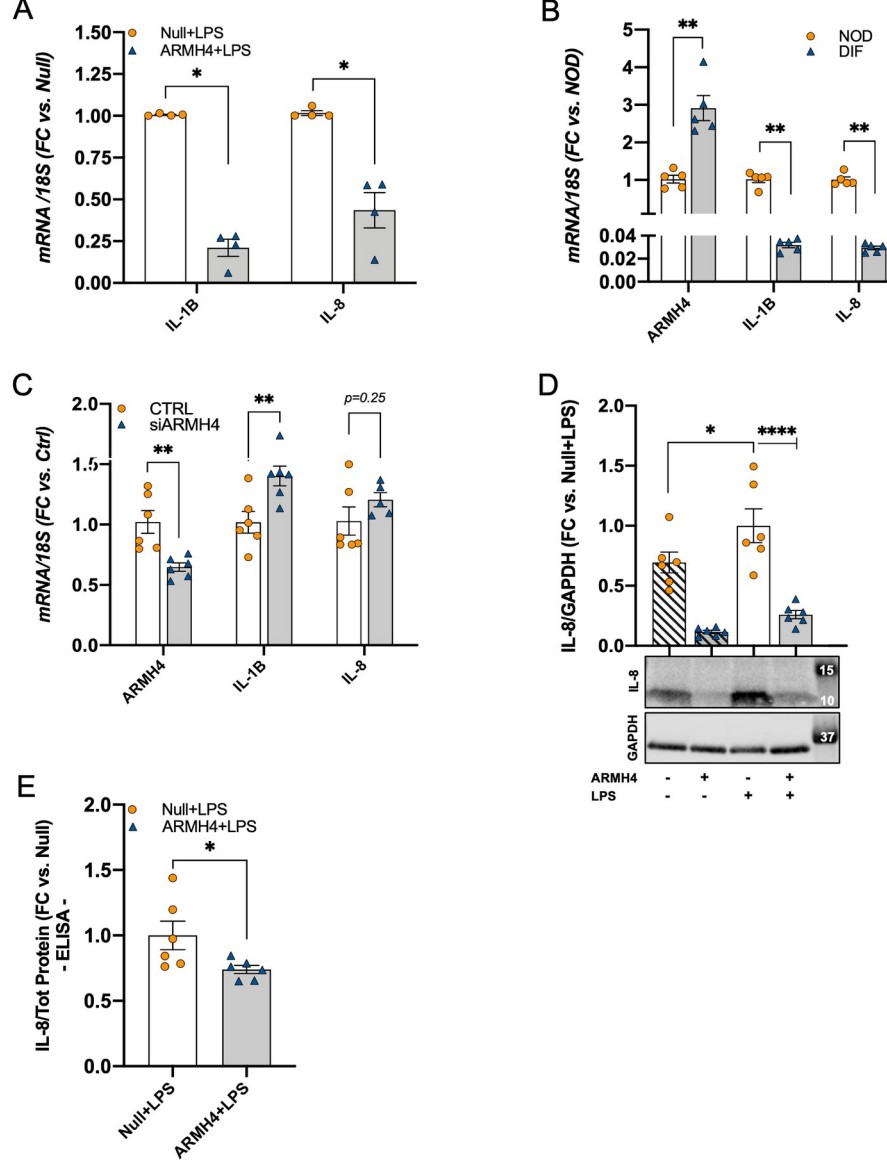

**Fig 4. ARMH4 may protect podocytes from inflammatory insults. A**: viral overexpression of ARMH4 reduces *IL-1B* and *IL-8* mRNA levels in hPTCs stimulated with LPS. **B**: the levels of both *IL-1B* and *IL-8* are reduced in differentiated podocytes in parallel with the spontaneous increases of ARMH4. **C**: silencing ARMH4 results in the upregulation of *IL-1B*. The anti-inflammatory effect of ARMH4 was confirmed for IL-8 at the protein level using Western blotting (**D**) and using ELISA (**E**). For these two latter experiments, samples are from two patients with three replicates from each ($n = 6$). Total protein concentration in cell lysates was used to normalize protein levels in supernatants.

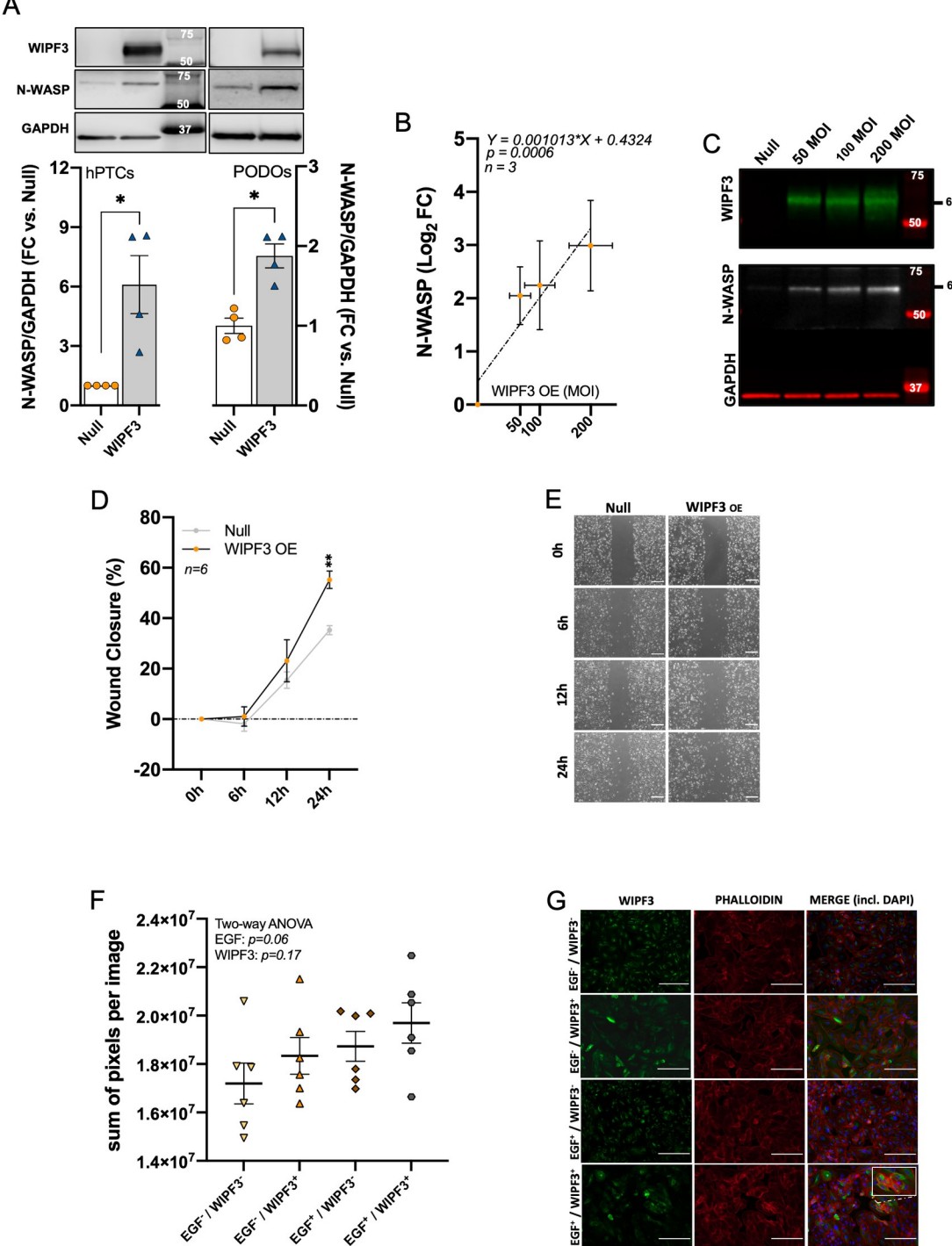

**Fig 5. WIPF3 may contribute to cell movement and cytoskeletal regulation in podocytes. A**: N-WASP protein levels increase in both hPTCs and podocytes after overexpression of WIPF3. **B**: stabilization of N-WASP increases linearly with WIPF3 overexpression (WIPF3 OE), and representative Western blots are shown in **C**. **D**: podocytes overexpressing WIPF3 show increased migration compared to null cells in the wound healing assay. Representative images of the scratch wound at different time points are shown in **E** (scale bars: 100 μm). **F**: phalloidin staining, an indication of actin filament formation, was assayed in hPTCs after overexpression of WIPF3 and treatment with the CDC42 activator EGF. The effects of EGF and WIPF3 were tested with a two-way ANOVA ($p = 0.06$ for EGF and $p = 0.17$ for WIPF3). Representative images are shown in **G** (scale bars: 100 μm). Cells from three patients, with two replicates from each, were analyzed ($n = 6$).

and 5C). To gauge a possible role of WIPF3 in cytoskeletal dynamics, we performed a wound healing assay in cells treated with Null and WIPF3 adenoviruses. Overexpression of WIPF3, from low endogenous levels, promoted wound closure (Fig 5D). Representative images of the gap closure for both groups at different time points are shown in Fig 5E. Finally, we used EGF to activate Cdc42 in hPTCs overexpressing WIPF3, followed by phalloidin staining of filamentous actin. Neither EGF treatment nor WIPF3 overexpression significantly affected actin cable formation, even though both tended to have an effect as shown in Fig 5F. Representative immunofluorescence images are shown in Fig 5G.

## The fate of ARMH4 and WIPF3 in podocytic disease

Having established that ARMH4 and WIPF3 are markers of healthy human podocytes, we next asked if they change in disease. For this, we used the NephroSeq database and examined kidney biopsies from the routine pathology service at our hospital. Glomerular gene profiles available in NephroSeq [28], revealed that both *ARMH4* (*C14orf37*) and *WIPF3* were reduced in diseased kidneys, with the largest difference seen for *ARMH4* in collapsing and classic focal segmental glomerulosclerosis (Coll. and Class. FSGS) versus normal kidney (NK) (Fig 6A and 6B). Moreover, in-house immunohistochemistry on renal biopsies from patients with glomerular diseases, where podocyte involvement is pathophysiologically central, showed apparent reduction in podocytic staining for ARMH4 vs. normal kidney (NK) (Fig 6C). Minimal change disease (MCD), focal segmental glomerulosclerosis (FSGS), membranous nephropathy (MN), and pauci-immune crescentic glomerulonephritis (CG) were investigated. In CG, in particular, ARMH4 was observed to be absent in the crescentic part of the affected glomerulus but still present in a healthy area (compare G1 and G2 subareas in Fig 6C, micrograph CG). To facilitate visualization of the crescent and normal portion of this glomerulus, images were obtained from two sections taken in sequential order. One of these was stained using H&E (CG-H&E) and the other one using immunohistochemistry (CG). In CG, ARMH4 was not exclusively seen in podocytes, but it was also evident in some cells of an enlarged and injured proximal tubule, as shown by arrows (Fig 6C).

## Discussion

This study demonstrates the feasibility of using a computational approach and publicly available bulk RNA-sequencing data combined with histology repositories to identify human podocyte markers. This workflow complements ongoing single cell transcriptomic characterization of kidney cell types [6,29], and, in theory, it has certain strengths. For instance, incompletely dissociated cell clusters in single cell RNA-seq may lead to artefactual assignment of properties to hybrid cell types. Moreover, the number of biological replicates in single cell RNA-seq studies is typically small in comparison to the 85 individuals used here.

In support of our bioinformatical findings, we used a battery of complementary methods to support podocyte localization of ARMH4 and WIPF3, which were among the podocyte markers identified. Both immunofluorescence and immunohistochemistry supported localization of these proteins in podocytes. Intriguingly, both imaging modalities suggested localization of ARMH4 to intracellular, perinuclear, structures. ARMH4 is predicted to be a membrane protein, and when we prepared subcellular fractions of cells in which ARMH4 was overexpressed, it was almost exclusively localized in the membrane fraction. Taken together, this suggest that, in podocytes, ARMH4 localizes to intracellular membranes, such as the endoplasmic reticulum or Golgi apparatus. Another peculiar finding regarding ARMH4 is that the predicted molecular weight is 84 kDa, but we found that overexpressed ARMH4 migrated at 130 kDa. A 130 kDa band was moreover enriched in the membrane fraction of human kidney cells

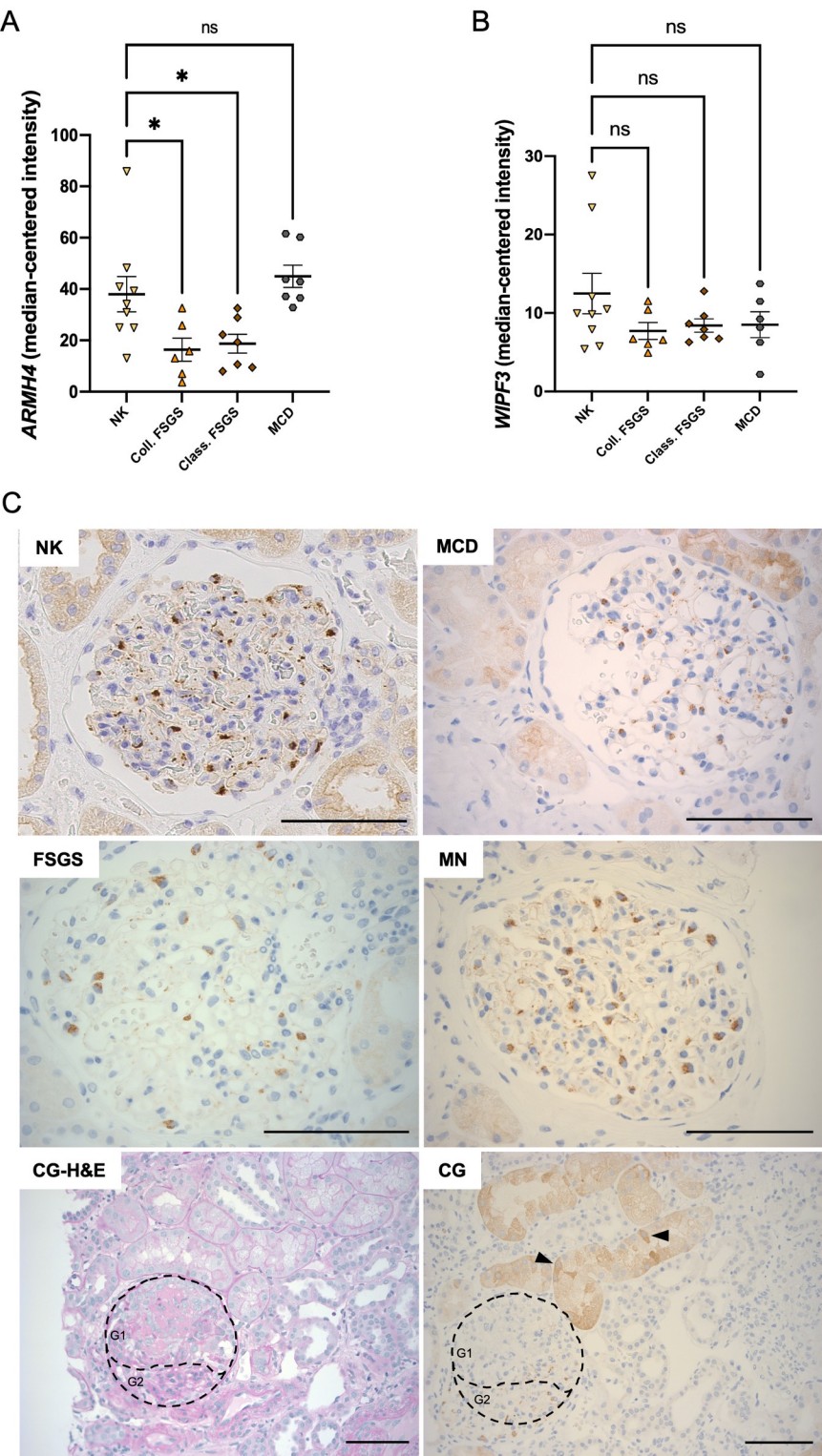

**Fig 6. ARMH4 and WIPF3 in the healthy and disease state. A**: the RNA level of *ARMH4* is reduced in patients suffering from collapsing (coll.) and classic (class.) FSGS compared to normal kidney (NK), but not in those diagnosed with minimal change disease (MCD). The *WIPF3* level, on the other hand, was not significantly different in any of the diseases (**B**). **C**: immunohistochemistry of glomeruli in patients diagnosed with MCD (*n* = 2), FSGS (*n* = 2), MN (*n* = 2) and CG (*n* = 2) suggests lower intensity of ARMH4 immunoreactivity vs. normal kidney (NK). In CG, ARMH4

is not seen in the crescent of a glomerulus while some staining is maintained in the healthy-looking part of the same glomerulus (G1 = crescent vs. G2 = healthy tissue). The glomerular and tubular morphology were supported by H&E staining (CG-H&E). Arrows in CG show ARMH4 in some cells of a swollen proximal tubule. Scale bars: 100 μm.

overexpressing ARMH4 as well as in glomeruli relative to the whole cortex. A 245 kDa band was also enriched in glomeruli, and we speculate that this heavy species arises due to multimerization or post-translational modifications. In addition to being enriched in podocytes, we found that ARMH4 increases during *in vitro* differentiation of podocytes, and, like WIPF3, it responds at the mRNA level to overexpression of the podocyte transcription factor LMX1B.

ARMH4 has been scarcely studied previously, but two studies from other organs indicated that this protein exerts an anti-inflammatory effect by reducing IL-6-dependent phosphorylation of AKT and STAT3 [23,24]. In keeping with this possibility, we observed clear-cut reductions of IL-1B and IL-8 secretion after overexpression of ARMH4 in hPTCs exposed to LPS. Moreover, in the podocyte differentiation paradigm, where ARMH4 increased, IL-1B and IL-8 were reduced. Silencing of ARMH4, finally, increased IL-1B. Such an immune modulatory and anti-inflammatory effect of this protein is of clear interest given the previously reported association between a variant at the ARMH4 locus and lupus nephritis [12]. It remains to be determined if specific molecular manipulation of this protein *in vivo* has pro-inflammatory consequences in the kidney.

For WIPF3, immunofluorescence staining and immunohistochemistry were again consistent, and podocyte localization was evident. This verprolin family member has been reported to form a complex with N-WASP, protecting it from degradation. This complex is important for actin nucleation and localizes to the tips of filopodia in branching cells [25,26]. Moreover, N-WASP is required to maintain podocyte foot processes [2]. We demonstrate here that overexpression of WIPF3 increases the N-WASP protein level in both tubular cells and podocytes, suggesting that WIPF3 may regulate actin dynamics. Overexpression of WIPF3 did indeed affect cell migration in the wound healing assay, and it tended to promote overall phalloidin staining in cell culture. The latter assay may however be too crude to measure Cdc42-dependent impact on the cytoskeleton because we did not see an effect with the positive control. Given the effect on cell migration, WIPF3 may nonetheless have a more subtle effect on actin dynamics via its stabilizing effect on N-WASP. Interestingly, *WIPF3* has been captured in genome-wide association studies for albuminuria, arguing that this protein plays a role for the barrier properties of podocytes.

Using available RNA-seq data from human glomeruli [28], *ARMH4* was found to be downregulated in glomerular diseases such as collapsing and classic focal segmental glomerulosclerosis (FSGS) compared to normal tissue. *WIPF3* on the other hand remained steadily expressed irrespective of the disease. In line with this, our independent immunohistochemistry on biopsies from patients suffering from minimal change disease, FSGS, membranous nephropathy, and pauci-immune crescentic nephritis showed an apparent downregulation of ARMH4 immunoreactivity. In crescentic glomerulonephritis, in particular, ARMH4 seemed to be normally expressed in the smaller healthy segment of a glomerulus, while it was absent in a fibrotic area of the same glomerulus. Given the anti-inflammatory impact of ARMH4, we speculate that its apparent reduction in kidney diseases may contribute to inflammation in glomerulonephritides.

To summarize, using a bioinformatical approach and complementary methods to support protein distribution, this work identifies ARMH4 and WIPF3 as proteins enriched in human

podocytes, possibly protecting these cells from inflammatory insults and contributing to their actin dynamics.

## Supporting information

**S1 File.**
(XLSX)

**S1 Raw images.**
(PDF)

## Acknowledgments

We would like to thank Catarina Rippe for tutorials in protein isolation, Gülay Altiparmak for immunohistochemistry and Roberto Boi for advice on podocyte culturing.

## Author Contributions

**Conceptualization:** Francesco De Luca, Karl Swärd, Martin E. Johansson.

**Data curation:** Francesco De Luca.

**Formal analysis:** Francesco De Luca.

**Funding acquisition:** Martin E. Johansson.

**Investigation:** Martin E. Johansson.

**Methodology:** Francesco De Luca, Michelle Kha, Karl Swärd, Martin E. Johansson.

**Project administration:** Karl Swärd, Martin E. Johansson.

**Supervision:** Karl Swärd, Martin E. Johansson.

**Writing – original draft:** Francesco De Luca, Karl Swärd, Martin E. Johansson.

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
