## [Decision Letter · Decision Letter 0]

19 Jul 2022

PONE-D-22-15620Identification of ARMH4 and WIPF3 as human podocyte proteins with potential roles in immunomodulation and cytoskeletal dynamicsPLOS ONE

Dear Dr. De Luca,

Thank you for submitting your manuscript to PLOS ONE. After careful consideration, we feel that it has merit but does not fully meet PLOS ONE’s publication criteria as it currently stands. Therefore, we invite you to submit a revised version of the manuscript that addresses the points raised during the review process.

We look forward to receiving your revised manuscript.

Kind regards,

Nicole Endlich, Prof

Academic Editor

PLOS ONE

Journal Requirements:

5. We note that you have included the phrase “not shown” in your manuscript. Unfortunately, this does not meet our data sharing requirements. PLOS does not permit references to inaccessible data. We require that authors provide all relevant data within the paper, Supporting Information files, or in an acceptable, public repository. Please add a citation to support this phrase or upload the data that corresponds with these findings to a stable repository (such as Figshare or Dryad) and provide and URLs, DOIs, or accession numbers that may be used to access these data. Or, if the data are not a core part of the research being presented in your study, we ask that you remove the phrase that refers to these data.

6. Please amend either the abstract on the online submission form (via Edit Submission) or the abstract in the manuscript so that they are identical.

Additional Editor Comments:

Dear Dr. de Luca,

The reviewers have sent their decisions. Because your manuscript contains many preliminary results that lack basic controls and have methodological deficiencies, I must inform you that a major revision is required.

Please address all objections.

Reviewers' comments:

Reviewer's Responses to Questions

**Comments to the Author**

1. Is the manuscript technically sound, and do the data support the conclusions?

Reviewer #1: Partly

Reviewer #2: Yes

2. Has the statistical analysis been performed appropriately and rigorously? 

Reviewer #1: Yes

Reviewer #2: Yes

3. Have the authors made all data underlying the findings in their manuscript fully available?

Reviewer #1: Yes

Reviewer #2: Yes

4. Is the manuscript presented in an intelligible fashion and written in standard English?

Reviewer #1: Yes

Reviewer #2: Yes

5. Review Comments to the Author

Reviewer #1: General:

In general, the approach applied by De Luca and colleagues is feasible to identify new podocyte markers which with little doubt is of great importance for the field. In general, the manuscript is well written, and most parts sufficiently discussed. However, especially the post-screening validation and their preliminary functional characterization lack fundamental controls and has methodological downfalls.

Major points:

1. The authors state in the discussion regarding the finding in Fig. 2: “Both immunofluorescence and immunohistochemistry supported localization of these proteins in podocytes, but the subcellular distribution differed for ARMH4 depending on the staining method.” For a new, yet undescribed protein suitable antibody controls are required here. The minimal requirement for a new protein of interest would be at least a control lacking the primary antibody (or better a non-specific antibody of the same isotype) or the same staining with primary antibody that has been preincubated with a blocking peptide. Otherwise, doubts will remain over specificity of the antibody, especially if localization differs between staining methods. A second, independent antibody (meaning not used in the HPA IHC pipeline that was a component of the screen) would strengthen this finding as well.

2. To conclude from viral overexpression in proximal tubule cells to a stereotypic cellular response in podocytes is far-fetched. How do cultured podocytes react upon viral transfection of the respective genes? To me, it is not clear why the researchers have not used the podocyte cell line for this as well. Cytoskeletal reorganization in specialized cells like podocytes differs substantially from proximal tubular cells. Therefore, to conclude to a cell-specific phenotype, these important investigations should be performed in podocytes.

3. The second part of the title is misleading. In line with major point 2: Immunomodulation is not described here, yet properly discussed. Doubtless, this is an important point for postmitotic podocytes, but the results presented here do not allow such conclusion. In line with that: Solely by the increase of N-WASP, the authors suggest a role in cytoskeletal dynamics. To conclude to a role in cytoskeletal dynamics more experiments are needed: A starter can be morphological assessment of actin cytoskeleton after transfection and/or functional assays like migration assays.

Minor points:

1. For the screening validation: The authors could integrate publicly available scRNA sequencing datasets in their analysis to strengthen their targets. A short look indeed supported their finding of podocyte specificity of at least WIPF3 which in several datasets clusters in the podocyte fraction. This is a chance for external validation of their findings. For mouse https://cello.shinyapps.io/kidneycellexplorer/ and for human: http://humphreyslab.com/SingleCell/

2. Figure 2: The general quality of the immunofluorescence micrographs is not convincing and very dim. It may be that during figure conversion the general fluorescence intensity was impaired.

3. Regarding the western blot results: From the methods it is not clear whether the authors used reducing or non-reducing conditions for the SDS-PAGE. The choice of the loading buffer can influence the height protein of interest will run in a gel. This could lead to such unforeseen results.

4. Figure 3 E: Quantitative image analysis should be used to demonstrate this upregulation (e.g. mean fluorescence

Reviewer #2: De Luca and colleques identified so far unknown podocyte proteins with potential roles in immunomodulation and cytoskeletal dynamics. New podocyte specific proteins were identified by using publicly available databases for RNAseq, the Human Protein Atlas and computational analysis. Hereby they identified potential novel podocyte markers. Two of them were selected for further analysis in immunofluorescence und immunohistochemistry as well as overexpression experiments in vitro. Transcripts of both genes (ARMH4 and WIPF3) was increased after overexpression of the podocyte transcription factor LMX1B. Furthermore, overexpression of ARMH4 in primary kidney epithelial cells reduced IL-1B and IL-8. In contrast, overexpression of WIPF3 stabilized N-WASP, which is required for maintenance of podocyte foot processes.

General comments:

This study is an interesting and important research study. However, there are some points harming the enthusiasm for this paper:

MM section

1. Despite RNA-Seq data are derived from an existing database, it would be helpful to get more information on samples that were analysed in this study, e.g. sample number, healthy or diseased patients?, It also remains unclear how many kidney tissues from tumor nephrectomies were analysed

2. Information for the used secondary antibody in immunohistochemistry is missing.

3. Using the protocol described for the isolation of primary proximal tubular cells I would expect that you will establish a culture of tubular cells with mixed origin, not restricted to proximal tubular cells.

4. The method used to identify new transcripts of podocytes worked in the present work. However, to my understanding, podocyte specific genes may be regulated differently than the genes used such as nephrin, podocin and Glepp1. In particular, when podocytes are damaged, slit membrane proteins are known to be downregulated. Podocyte-specific injury markers, however, could then be upregulated, for example, and not correlate with this. The elucidation of such podocyte-specific proteins would also contribute to a better understanding of pathological changes.

Results section

5. Overall, the illustration for this publication is in poor resolution. Therefore, figures and labels appear pixelated. In the immunofluorescence staining it is impossible to recognize the described staining. Here, the fluorescence signal should be enhanced. The use of confocal microscopy could also be very helpful here and verify co-localization with known podocyte proteins.

6. If I understood the authors correctly, there is no perinuclear inheritance in frozen section inheritance. What controls have been done to rule out non-specific inheritance in immunofluorescence or immunohistology? It is possible that immunohistochemical staining detects ARMH4 in endoplasmic reticulum membranes. However, it is unclear why this is not seen in the frozen sections. Perhaps different methods should be tested for the antigen retrieval.

7. Furthermore, it remained unclear to me why tubule cells and not podocytes were used for the overexpression experiments.

Discussion

8. Is there any idea whether ARMH4 fulfills its function as a urate transporter and what is the importance of this for the podocyte?

9. It would also be interesting to investigate the significance of the two newly described podocyte proteins in renal diseases. Suitable for this would be either immunohistological studies on biopsy material from patients with different human kidney diseases. Another possibility to get an impression whether these proteins are up- or down-regulated during renal disease would be the analysis of expression data accessible in the NephroSeq database. Including such data would significantly increase the relevance of the manuscript.

6. PLOS authors have the option to publish the peer review history of their article (what does this mean?). If published, this will include your full peer review and any attached files.

Reviewer #1: No

Reviewer #2: **Yes: **Christoph Daniel

---

## [Author Response · Author response to Decision Letter 0]

19 Oct 2022

EDITOR:

Response: We have carefully checked the formatting requirements and made changes. We hope that the manuscript now complies with all the requirements. 

2. Please provide additional details regarding participant consent. In the ethics statement in the Methods and online submission information, please ensure that you have specified what type you obtained (for instance, https://journals.plos.org/plosone/s/file?id=wjVg/PLOSOne_formatting_sample_main_body.pdf and 

written or verbal, and if verbal, how it was documented and witnessed). If your study included minors, state whether you obtained consent from parents or guardians. If the need for consent was waived by the ethics committee, please include this information.

Response: Written informed consent was obtained. This is now clearly stated in the materials and methods section of the manuscript. 

Response: We have edited these sections so that they match. 

4. PLOS ONE now requires that authors provide the original uncropped and unadjusted images underlying all blot or gel results reported in a submission’s figures or Supporting Information files. 

When you submit your revised manuscript, please ensure that your figures adhere fully to these guidelines and provide the original underlying images for all blot or gel data reported in your submission. 

Response: We have provided uncropped blots among the supporting information files. This is indicated in the cover letter.

5. We note that you have included the phrase “not shown” in your manuscript. Unfortunately, this does not meet our data sharing requirements. PLOS does not permit references to inaccessible data. We require that authors provide all relevant data within the paper, Supporting Information files, or in an acceptable, public repository. Please add a citation to support this phrase or upload the data that corresponds with these findings to a stable repository (such as Figshare or Dryad) and provide and URLs, DOIs, or accession numbers that may be used to access these data. Or, if the data are not a core part of the research being presented in your study, we ask that you remove the phrase that refers to these data.

Response: Phrase removed, and any additional data added to supplementary files.

6. Please amend either the abstract on the online submission form (via Edit Submission) or the abstract in the manuscript so that they are identical.

Response: Changed as requested.

REVIEWER #1: 

General:

In general, the approach applied by De Luca and colleagues is feasible to identify new podocyte markers which with little doubt is of great importance for the field. In general, the manuscript is well written, and most parts sufficiently discussed. However, especially the post-screening validation and their preliminary functional characterization lack fundamental controls and has methodological downfalls.

Major points:

1. The authors state in the discussion regarding the finding in Fig. 2: “Both immunofluorescence and immunohistochemistry supported localization of these proteins in podocytes, but the subcellular distribution differed for ARMH4 depending on the staining method.” For a new, yet undescribed protein suitable antibody controls are required here. The minimal requirement for a new protein of interest would be at least a control lacking the primary antibody (or better a non-specific antibody of the same isotype) or the same staining with primary antibody that has been preincubated with a blocking peptide. Otherwise, doubts will remain over specificity of the antibody, especially if localization differs between staining methods. A second, independent antibody (meaning not used in the HPA IHC pipeline that was a component of the screen) would strengthen this finding as well.

Response: We agree with the reviewer. Primary antibody omission controls were negative and control micrographs have now been included in the figure to illustrate this. Regarding antibodies, a thorough search for ARMH4 antibodies seemingly identifies about a dozen of potential antisera usable for immunohistochemistry. However, they all emanate from two distinct sera, sold under different brand names. We have now tested all antibodies recommended for ARMH4 imaging and the different antibodies give the same results. 

We were concerned ourselves that there might be a bleedthrough/cross-reaction in the co-staining of ARMH4 and NPHS2 in our original IF images. This was discussed in the original manuscript. The inconsistency between IF and IHC was now addressed with new IF stainings using a different protocol. Critical for this new imaging was identification of a mouse anti-NPHS2 antibody allowing us to skip the conjugation step used for one of two rabbit antibodies. When repeating the original IF double stainings we get results that are consistent with the old IHC-stainings, showing a perinuclear localization of ARMH4 in podocytes with little or now staining in the plasma membrane. 

2. To conclude from viral overexpression in proximal tubule cells to a stereotypic cellular response in podocytes is far-fetched. How do cultured podocytes react upon viral transfection of the respective genes? To me, it is not clear why the researchers have not used the podocyte cell line for this as well. Cytoskeletal reorganization in specialized cells like podocytes differs substantially from proximal tubular cells. Therefore, to conclude to a cell-specific phenotype, these important investigations should be performed in podocytes.

Response: The bulk of experiments in the original version of the manuscript were done using proximal tubular cells because we were unable at the time to allocate a cell culture incubator to the lower temperature (required for podocyte proliferation). We have now obtained new batches of immortalized podocytes and redone most transduction experiments. We provide new data on the effects of podocyte differentiation, showing that ARMH4 mRNA levels increase with podocyte differentiation (new fig 4B), whereas IL1B and IL8 mRNA levels decline. In addition we show effects of ARMH4 overexpression and siRNA mediated ARMH4 knockdown in relation to IL1B and IL8 (new fig 4C, D). For WIPF3 overexpression, we found similar stabilization of N-WASP (WASL) in podocytes as as in proximal tubular cells (data now included in Figure 5A).

For overexpression experiments, it is an advantage if the gene product of interest has low expression under basal conditions. Differentiated podocytes spontaneously express ARMH4 so any ARMH4 effect may already be saturated. It therefore makes sense to keep the data on proximal tubular cells in the manuscript.

3. The second part of the title is misleading. In line with major point 2: Immunomodulation is not described here, yet properly discussed. Doubtless, this is an important point for postmitotic podocytes, but the results presented here do not allow such conclusion. In line with that: Solely by the increase of N-WASP, the authors suggest a role in cytoskeletal dynamics. To conclude to a role in cytoskeletal dynamics more experiments are needed: A starter can be morphological assessment of actin cytoskeleton after transfection and/or functional assays like migration assays.

Response: We thank the reviewer for this suggestion. First, we have moderated our discussion on immunomodulation in order not to overstretch our conclusions. Furthermore, as suggested by the reviewer, we have performed a migration (scratch) assay. It shows that WIPF3 overexpression increases the migratory capacity of the transfected cells (new fig 5 D, E). Because prior studies had suggested that verprolin family proteins (e.g., WIPF3) contribute to formation of microspikes, small membrane protrusions reminiscent of minor foot processes, we also tried to measure such structures, but we did not see an effect. Instead, and as also suggested by the reviewer, we imaged the actin cytoskeleton following WIPF3 transfection. These results have been added to the manuscript (new figure 5F, G).

Minor points:

1. For the screening validation: The authors could integrate publicly available scRNA sequencing datasets in their analysis to strengthen their targets. A short look indeed supported their finding of podocyte specificity of at least WIPF3 which in several datasets clusters in the podocyte fraction. This is a chance for external validation of their findings. For mouse https://cello.shinyapps.io/kidneycellexplorer/ and for human: http://humphreyslab.com/SingleCell/

Response: We have acted according to the suggestion. Confirmation in independent scRNA datasets has been indicated in the results section, but we have chosen https://patrakkalab.se/kidney/ for human. According to this database, ARMH4 (typing the alias C14orf37) showed clearly enrichment in human podocytes. When searching ARMH4 (or C14orf37) in the databases from Humphreys lab the gene was not found. 

2. Figure 2: The general quality of the immunofluorescence micrographs is not convincing and very dim. It may be that during figure conversion the general fluorescence intensity was impaired.

Response: It may have been an issue with the conversion. We have now provided images with the highest resolution attainable and hope that the images are satisfactory.

3. Regarding the western blot results: From the methods it is not clear whether the authors used reducing or non-reducing conditions for the SDS-PAGE. The choice of the loading buffer can influence the height protein of interest will run in a gel. This could lead to such unforeseen results.

Response: We use reducing conditions for western blotting. This is now more clearly indicated. For the fractionation experiments we used the buffer provided with the kit. Samples were further denatured with SDS prior to protein determination.

4. Figure 3 E: Quantitative image analysis should be used to demonstrate this upregulation (e.g., mean fluorescence).

Response: Quantitative image analysis has been performed and results have been added to the manuscript. 

REVIEWER #2: 

De Luca and colleagues identified so far unknown podocyte proteins with potential roles in immunomodulation and cytoskeletal dynamics. New podocyte specific proteins were identified by using publicly available databases for RNAseq, the Human Protein Atlas and computational analysis. Hereby they identified potential novel podocyte markers. Two of them were selected for further analysis in immunofluorescence und immunohistochemistry as well as overexpression experiments in vitro. Transcripts of both genes (ARMH4 and WIPF3) was increased after overexpression of the podocyte transcription factor LMX1B. Furthermore, overexpression of ARMH4 in primary kidney epithelial cells reduced IL-1B and IL-8. In contrast, overexpression of WIPF3 stabilized N-WASP, which is required for maintenance of podocyte foot processes.

General comments:

This study is an interesting and important research study. However, there are some points harming the enthusiasm for this paper:

MM section

1. Despite RNA-Seq data are derived from an existing database, it would be helpful to get more information on samples that were analysed in this study, e.g. sample number, healthy or diseased patients? It also remains unclear how many kidney tissues from tumor nephrectomies were analysed.

Response: There are kidney cortices from 85 individuals in the GTExPortal and harvested tissues are grossly normal. This is now clearly stated in M&M. For in-house staining of healthy kidney tissue we used kidneys from nephrectomies due to localized kidney cancer. Healthy kidney tissue from 4 individuals was used. 

2. Information for the used secondary antibody in immunohistochemistry is missing.

Response: We thank the reviewer for pointing this out. We use the Envision IHC kit from Dako/Agilent, where the secondary antibody is included in the kit. Information regarding this kit has now been provided.

3. Using the protocol described for the isolation of primary proximal tubular cells I would expect that you will establish a culture of tubular cells with mixed origin, not restricted to proximal tubular cells.

Response: We are using well established protocols and we have performed extensive characterizations of the cultures over the years. There is no overgrowth of fibroblasts, endothelial cells or other cells than proximal tubular cells. These attain a somewhat altered marker expression however. We have compared the primary cells from our lab with commercially available primary cells, without identifying a difference.

4. The method used to identify new transcripts of podocytes worked in the present work. However, to my understanding, podocyte specific genes may be regulated differently than the genes used such as nephrin, podocin and Glepp1. In particular, when podocytes are damaged, slit membrane proteins are known to be downregulated. Podocyte-specific injury markers, however, could then be upregulated, for example, and not correlate with this. The elucidation of such podocyte-specific proteins would also contribute to a better understanding of pathological changes.

Response: This is a very good point, but GTExPortal only presents RNA-seq data from tissues that are grossly normal. We have therefore extracted data from the NephroSeq database for inclusion in the manuscript. Results show a modest downregulation of ARMH4 in focal segmental glomerusclerosis. In view of this finding we also performed in-house immunohistochemistry on kidney biopsies from patients with different kidney diseases. Although limited in number, we find that the staining intensity for ARMH4 in the samples is reduced, especially in crescentic glomerulnephritis. The new data was added in a new figure 6.

Results section

5. Overall, the illustration for this publication is in poor resolution. Therefore, figures and labels appear pixelated. In the immunofluorescence staining it is impossible to recognize the described staining. Here, the fluorescence signal should be enhanced. The use of confocal microscopy could also be very helpful here and verify co-localization with known podocyte proteins.

Response: We believe this may have been an issue with the PDF conversion. We have now provided images with higher resolution.

6. If I understood the authors correctly, there is no perinuclear inheritance in frozen section inheritance. What controls have been done to rule out non-specific inheritance in immunofluorescence or immunohistology? It is possible that immunohistochemical staining detects ARMH4 in endoplasmic reticulum membranes. However, it is unclear why this is not seen in the frozen sections. Perhaps different methods should be tested for the antigen retrieval.

Response: Your interpretation is correct. Our initial IF stainings suggested that ARMH4 is localized to the plasma membrane while immunohistochemistry results suggested an intracellular perinuclear localization. We were concerned ourselves that there might be bleedthrough/cross-reaction during the co-staning ARMH4 and NPHS2 in our original IF images because we covalently labeled the NPHS2 antibody using a kit. This was therefore discussed in the original MS. We have now identified a mouse anti-NPHS2 antibody, which made it possible repeat the IF with sera from two different species. The new IF images are consistent with the IHC-results, showing perinuclear staining of ARMH4 in podocytes with little or no staining in the external plasma membranes. We apologize for our original error. 

7. Furthermore, it remained unclear to me why tubule cells and not podocytes were used for the overexpression experiments.

Response: The scientific rationale was that ARMH4 increases spontaneously as podocytes differentiate and we wanted cells devoid of ARMH4 as controls to preclude spontaneous saturation of any effect that ARMH4 may have. It was also a resource problem as we did not have an incubator that we could regularly set to the different temperature for a long period of time. 

Still, podocytes are the cells of interest and we therefore obtained new batches of immortalized podocytes for this revision.We have now repeated key experiments with podocytes and added several new panels in many figures. In the new Fig 3G and H we present data showing that ARMH4 protein as well as N-WASP protein increase with differentiation. In Fig 4B we also present new podocyte data regarding the inverse relation between ARMH4 and IL1B and IL8 levels. In 4C we use siRNA in podocytes for ARMH4 knockdown and find that IL1B increases. In Fig 5A, N-WASP data after WIPF3 overexpression in podocytes was added (shown to the right of the old data on tubular epithelial cells). WIPF3 stabilizes N-WASP in both cell types.

Discussion

8. Is there any idea whether ARMH4 fulfills its function as a urate transporter and what is the importance of this for the podocyte?

Response: We are unaware of any studies suggesting that ARMH4 is a urate transporter, and to the best of our knowledge, it does not share significant homology to known urate transporters. Can you provide a reference please?

9. It would also be interesting to investigate the significance of the two newly described podocyte proteins in renal diseases. Suitable for this would be either immunohistological studies on biopsy material from patients with different human kidney diseases. Another possibility to get an impression whether these proteins are up- or down-regulated during renal disease would be the analysis of expression data accessible in the NephroSeq database. Including such data would significantly increase the relevance of the manuscript.

Response: Excellent suggestion! We have now carried out immunohistochemical stainings in a small set of podocytic glomerular diseases. ARMH4 staining seems to decrease somewhat in the podocytic diseases. Most dramatically so in the crescents of crescentic glomerulonephritis, where expression is completely abolished. Maybe somewhat surprisingly, ARMH4 also seems to become expressed de-novo in swollen proximal tubules. The new data was included in a new figure 6. We also included expression data from the NephroSeq database. Our in-house staining in FSGS seems to corroborate the ARMH4 expression data from Nephroseq.

In conclusion, we believe that the constructive criticism and suggestions provided by the reviewers have resulted in a considerably improved manuscript. We hope that the measures taken are adequate and meet the expectations of the editor and the reviewers. We wish to extend our gratitude to all involved parties for their work regarding the thorough review of this manuscript.

---

## [Decision Letter · Decision Letter 1]

26 Dec 2022

Identification of ARMH4 and WIPF3 as human podocyte proteins with potential roles in immunomodulation and cytoskeletal dynamics

PONE-D-22-15620R1

Dear Mr. De Luca,  

We’re pleased to inform you that your manuscript has been judged scientifically suitable for publication and will be formally accepted for publication once it meets all outstanding technical requirements.

Kind regards,

Peter R. Corridon

Academic Editor

PLOS ONE

Additional Editor Comments (optional):

The authors have satisfactorily addressed the reviewers' comments. I will ask the authors to do the following prior to publication:

- Add scale bars to several figures: e.g., Fig 3, 4; and

- Revise the level of fluorescence, as outlined by the reviewer. The current level still seems very weak, so it can only be seen reasonably well when the figure is enlarged to more than 100%. The intensities could be better emphasized with image processing of the individual fluorescences for printing the manuscript.

Reviewers' comments:

Reviewer's Responses to Questions

**Comments to the Author**

1. If the authors have adequately addressed your comments raised in a previous round of review and you feel that this manuscript is now acceptable for publication, you may indicate that here to bypass the “Comments to the Author” section, enter your conflict of interest statement in the “Confidential to Editor” section, and submit your "Accept" recommendation.

Reviewer #1: All comments have been addressed

Reviewer #2: All comments have been addressed

2. Is the manuscript technically sound, and do the data support the conclusions?

Reviewer #1: Yes

Reviewer #2: Yes

3. Has the statistical analysis been performed appropriately and rigorously? 

Reviewer #1: Yes

Reviewer #2: Yes

4. Have the authors made all data underlying the findings in their manuscript fully available?

Reviewer #1: Yes

Reviewer #2: Yes

5. Is the manuscript presented in an intelligible fashion and written in standard English?

Reviewer #1: Yes

Reviewer #2: Yes

6. Review Comments to the Author

Reviewer #1: With their revision, De Luca et al. have addressed all my major points. For example with the addition of appropriate control experiments, verification of findings in a podocyte cell line and a scratch assay in podocytes the paper has been substantially approved.

From my perspective, only one minor point has been left: Several micrographs are missing scale bars: e.g. Fig 3, 4

Reviewer #2: The authors have satisfactorily answered the questions I asked and made appropriate changes and additions to the manuscript. However, the fluorescence still seems very weak to me, so that it can only be seen reasonably well when the figure is enlarged to more than 100%. Perhaps the intensities could be better emphasised with image processing of the individual fluorescences for printing the manuscript.

PS: Unfortunately, I can no longer find a citation for my claim that ARMH4 is a urate transporter; I must have made a mistake in my research, so please excuse me for this mishap.

7. PLOS authors have the option to publish the peer review history of their article (what does this mean?). If published, this will include your full peer review and any attached files.

Reviewer #1: No

Reviewer #2: **Yes: **Christoph Daniel

---

## [Editor Report · Acceptance letter]

4 Jan 2023

PONE-D-22-15620R1 

Identification of ARMH4 and WIPF3 as human podocyte proteins with potential roles in immunomodulation and cytoskeletal dynamics 

Dear Dr. De Luca:

I'm pleased to inform you that your manuscript has been deemed suitable for publication in PLOS ONE. Congratulations! Your manuscript is now with our production department. 

Kind regards, 

on behalf of

Dr. Peter R. Corridon 

Academic Editor

PLOS ONE